# Characterization of Current Husbandry and Veterinary Care Practices of the Giant Pacific Octopus (*Enteroctopus dofleini*) Using an Online Survey

**DOI:** 10.3390/vetsci10070448

**Published:** 2023-07-08

**Authors:** Ashley J. Kirby, Julie A. Balko, Caroline E. C. Goertz, Gregory A. Lewbart

**Affiliations:** 1Alaska SeaLife Center, Seward, AK 99664, USA; 2Department of Molecular Biomedical Sciences, North Carolina State University, College of Veterinary Medicine, Raleigh, NC 27607, USA; 3Department of Clinical Sciences, North Carolina State University, College of Veterinary Medicine, Raleigh, NC 27607, USA

**Keywords:** octopus, anesthesia, sedation, veterinary

## Abstract

**Simple Summary:**

Giant Pacific octopuses (*Enteroctopus dofleini*) (GPOs) are commonly found in zoos or aquariums. However, veterinarians or animal care professionals may be reluctant to perform diagnostic or other procedures due to limited information on best practices for sedation, anesthesia, or euthanasia of the species. The purpose of this study was to survey aquatic veterinarians and animal care professionals on current animal and veterinary care, including anesthesia and euthanasia, of GPOs. An online survey was distributed with participation from 52 institutions. Results included animal care practices consistent with current recommendations and highly variable involvement with veterinary care. Over 20 institutions have used anesthesia or sedation for procedures that do not involve euthanasia. Anesthesia and euthanasia methods were reported, including a variety of techniques and large dose ranges compared to techniques used for smaller cephalopods. Observations described include side effects such as failure to adequately anesthetize or euthanize, inking, delayed recovery, or behavior changes. The results of this study may help guide additional studies in GPO anesthesia.

**Abstract:**

Giant Pacific octopuses (*Enteroctopus dofleini*) (GPOs) are commonly housed in zoos or aquaria, and sedation, anesthesia, and/or euthanasia may be indicated for a variety of reasons. Despite this need, evidence-based data on best practices is limited and focuses on smaller or more tropical species. The objectives of this study were to survey the aquatic community regarding the husbandry and veterinary care of GPOs, with a specific focus on anesthetic and euthanasia protocols. A two-part web-based survey was distributed to four aquatic and/or veterinary email listservs. Individuals from fifty-two institutions participated in phase one. Results documented that 40 (78 percent) participating institutions currently house GPOs, with most housing one and nine institutions housing two to three GPOs. The median (range) habitat volume is 5405 (1893–16,465) L, and 78 percent of systems are closed. Of the institutions surveyed, 23 have anesthetized or sedated a GPO for nonterminal procedures, including wound care, biopsies, and hemolymph collection. Reported methods of sedation or anesthesia include magnesium chloride, ethanol, isoflurane, tricaine methanesulfonate (MS-222), magnesium sulfate, benzocaine, and dexmedetomidine. Drugs or methods used for euthanasia include magnesium chloride, ethanol, mechanical decerebration, pentobarbital, isoflurane, MS-222, magnesium sulfate, benzocaine, potassium chloride, dexmedetomidine, and freezing. Reported observed side effects include ineffectiveness or inadequate sedation, inking, prolonged drug effects, and behavior changes. Survey data have the potential to guide the husbandry and veterinary care of GPOs and build the framework for future prospective studies on GPO sedation and anesthesia.

## 1. Introduction

Giant Pacific octopuses (*Enteroctopus dofleini*) are cephalopod molluscs commonly housed in zoological parks, aquaria, and research facilities. *E. dofleini* (GPOs) are the largest octopus species, with a habitat range including Pacific waters from Japan, Alaska and Southern California, USA [1]. GPOs are popular exhibit animals in managed care as a result of their unique anatomy, intelligence, and use in interactive educational programs [1]. Due to behaviors that demonstrate intelligence, interest in the welfare of cephalopods, including investigations of sentience, has increased, and preliminary guidelines for their care have been created [2,3]. More specifically, Fiorito et al. (2015) established guidelines for the husbandry and welfare of cephalopods used for research [4]. Additionally, the Association of Zoos and Aquariums (AZA) Giant Pacific Octopus (*Enteroctopus dofleini*) Care Manual provides general recommendations on environmental parameters, habitat design, transport, nutrition, reproduction, behavior, and veterinary management of GPOs in managed care [1]. Environmental conditions or husbandry factors can affect animal health and response to anesthesia. The results of a 2012 survey of 33 institutions on GPO husbandry have been reported in the AZA Giant Pacific Octopus (*Enteroctopus dofleini*) Care Manual [1]. While GPOs are commonly managed under human care, veterinary literature specific to the species is limited; one area in particular that is currently lacking is knowledge of best practices for sedation, anesthesia, and euthanasia.

While mortality can often be associated with senescence, disease prior to senescence is also reported, and further diagnostic workup may be warranted. Cephalopods are short-lived due to their semelparous life cycle of a single reproductive episode prior to death [5]. Clinical signs of senescence can be similar to other signs of disease in cephalopods, indicating diagnostic workup may be warranted in non-senescent individuals. Seeley et al. conducted a retrospective review of 19 aquarium GPO mortalities, of which only 10 (52.6 percent) were determined to be sexually mature and senescent at the time of death [6]. Inflammation of the gills (branchitis) was observed histologically in 14 individuals, of which protozoa consistent with *Ichthyobodo* species were found in 10 [6]. Moderate to severe branchitis was associated with these flagellates in nine GPOs, which indicates that gill biopsies may play a role in GPO monitoring for these parasites [6]. Sedation or anesthesia is warranted for diagnostics such as gill biopsies and hemolymph collection.

Prospective studies on sedation, anesthesia, and euthanasia are needed, but knowledge of current practices in other cephalopod species can help guide those studies. Anesthesia or sedation may be indicated for a variety of reasons, such as euthanasia, diagnostics, therapeutics, or transport. Due to the large role cephalopods play in research, literature is available on various anesthetic techniques [7,8,9,10,11,12,13]. One study established that magnesium chloride and ethanol both block afferent and efferent neural signals in the pallial nerve of cuttlefish (*Sepia bandensis*) and octopus (*Abdopus aculeatus, Octopus bocki*) species [14]. Hypothermia is also used as an immobilization technique in cephalopods, including GPOs [11,13,15]. The literature on anesthesia or euthanasia techniques specific to GPOs is limited. GPO exposure to ethanol at effective 2.5–3.3 percent doses has been reported [9,16]. Barord and Christie described the euthanasia of two senescent GPOs, which established doses of benzocaine that were effective for anesthesia and euthanasia [17]. While ethanol is a common and preferred anesthetic for many cephalopod species, challenges with GPOs have been reported, such as incomplete induction [9]. Due to presumed challenges and limited literature on other techniques, veterinarians may be less likely to recommend anesthesia for GPOs.

The objectives of this study were to survey individuals from the aquatic community regarding the husbandry and veterinary care of GPOs, including the use of anesthesia and protocol preferences.

## 2. Materials and Methods

A two-part online survey was created through Qualtrics^®^XM and approved by the North Carolina State University Institutional Review Board (study 22248). The voluntary survey was distributed through email and included a link to the online survey. Participants were required to confirm that they were over 18 years of age to proceed with the survey, but all other questions were optional. The question style included multiple-choice and open-ended questions.

The first survey was distributed to four listservs, including the International Association of Aquatic Animal Medicine (IAAAM), the American Association of Zoo Veterinarians (AAZV), the World Aquatic Veterinary Medical Association (WAVMA), and the Thelist@aquaticinfo.org listserv. The first survey (Appendix A) was open to any North American institution that currently or historically has housed cephalopods. To avoid duplication of results from an institution and to confirm that the participant was from a North American institution, participants were asked to provide the name of their institution. The 12 survey questions included whether the institution currently houses or has housed GPOs in the last five years and if a GPO has been sedated or anesthetized for any reason at the institution. Husbandry questions such as the size of the GPO habitat, average water temperature, and type of water system were also included. If a respondent answered “yes” to whether they currently house or have housed GPOs in the last five years, they were asked if they would be interested in participating in a second, more in-depth survey. If interested, these respondents had the option to provide email contact information. If multiple responses from an institution were received along with interest in the second survey, the respondents were notified with the provided contact information and asked to collaborate and submit one response in the second survey. The first survey was distributed 11 November 2020 through 16 November 2020, and responses were received 11 November 2020 through 21 April 2021.

The second survey was distributed to 35 respondents. This survey was also voluntary and included up to 23 branching questions on GPO husbandry, veterinary care, sedation, anesthesia, and euthanasia techniques. Question styles included multiple choice, check boxes, short answers, and open-ended. The second survey was distributed 24 November 2020 through 14 December 2020, and responses were received 25 November 2020 through 19 January 2021. Data for both phases were collected and reported as averages, medians, modes, and percentages.

## 3. Results

### 3.1. Survey One

Fifty-two institutions participated in the first survey, which consisted of 108 individual responses. Of those, 33 respondents only answered the first question, confirming their age of 18 or above. Nine responses could not be linked to a North American institution, and 14 individual responses were from an institution that had previously completed the survey.

Of the participating institutions, at the time of the survey, 40 (78 percent) currently housed GPOs, 11 did not currently house GPOs, and one respondent did not complete the question. Respondents that did not currently house GPOs were asked two follow-up questions. First, these respondents were asked if their institution planned to house GPOs in the future, and of the 11 responses, two selected “yes,” six selected “unsure,” and three selected “no.” Second, respondents were also asked if their institution had ever housed GPOs, and of the 11 responses, five institutions had never housed GPOs, one institution had housed GPOs but not in the last five years, and five institutions had housed GPOs in the last five years. For respondents from institutions that had never housed GPOs or had not housed GPOs in the last five years, the survey ended. Additional questions were asked of respondents from institutions that currently house GPOs or have housed GPOs within the past five years.

Respondents of institutions that currently housed GPOs at the time of the survey were asked how many GPOs are currently housed in a short answer question. Of the 30 responses, 21 (70 percent) institutions house one GPO, eight (27 percent) institutions house two GPOs, one institution (three percent) houses three GPOs, and three institutions house no GPOs. Note that the latter answer contradicts the preceding qualification for this question: the institution currently houses GPOs. A summary of responses to life support and water quality parameter questions is listed in Table 1. Respondents were next asked, “Has a GPO at your institution been sedated or anesthetized for any reason (including euthanasia)?” 23 (62 percent) selected yes, and 14 (38 percent) answered no.

Finally, respondents from institutions that currently housed or had housed GPOs in the last five years were asked to participate in a voluntary second, more in-depth survey. If these respondents answered “yes” (*n* = 35, 83 percent), they were prompted to provide their name and contact email address or other preferred contact method.

### 3.2. Survey Two

Of the 35 online survey invitations, there were 25 responses. When asked, “What is the source for GPOs at your facility?” with multiple answers accepted, 20 out of 22 respondents reported that the institution purchases GPOs commercially from vendors or receives donations from collectors, including bycatch. Four respondents reported GPOs are collected directly from the wild by the institution, two reported the institution obtains GPOs through facility transfer, and no respondents selected “other.” If an institution acquires GPOs as donations from an accidental catch, respondents were given space to elaborate on how often this may occur and provide additional details. Five respondents described the acquisition of animals from bycatch with frequency varying from rare (less than once a year) to up to six per year. Bycatch from hooks and crab traps was described.

Respondents were given open-ended questions on husbandry, including diet and water quality parameters. When asked, on average, how often a GPO at your institution is fed, 18 responses varied in frequency from twice daily to twice a week. Ten (56 percent) of respondents indicated that a GPO at their institution is fed three to five times per week. Sixteen respondents described the amount of food fed in terms of percentage of GPO body weight (one to five percent; *n* = 5), calories per weight of GPO (12.69 kcal/kg; *n* = 1), weight of food fed (80–338 g fed per feed; *n* = 6), amount per food type (whole vs. half of item; *n* = 1), and non-specific responses (*n* = 3). Types of teleost fish and invertebrate food items were listed, with the most popular responses including clams (14), shrimp (14), squid (12), capelin (12), and crab (11). Two respondents specified specific food items to be frozen and thawed, and three specified that the GPO might be offered live food. Enrichment feeds were described, including the use of Kong^®^ devices and frozen food popsicles. A summary of responses to life support and water quality parameter questions is listed in Table 1. Twenty respondents described life support system filtration methods and enclosure decor or materials, which are listed in Table 2.

Respondents were then asked a series of questions about veterinary care. Of the 10 (48 percent) out of 21 that responded “yes” regarding quarantine upon entrance, six respondents noted the length is typically 30 days. Three respondents noted a quarantine period of 14 to 28 days, and one respondent noted a quarantine period of 45 to 60 or more days. Responses to questions about the frequency and indications of veterinary examinations are listed in Table 3 and Table 4, with most only doing them as part of entrance examinations and at necropsy. When asked if any procedures excluding euthanasia have been performed on the facility’s GPOs in the last two years, eight (38 percent) respondents selected “yes,” thirteen (62 percent) respondents answered “no,” and none selected “unsure.’’ Respondents that selected “yes” were prompted to briefly list the procedures, which included biopsies (*n* = 6; including gills and skin), hemolymph collection (*n* = 2), skin scraping (*n* = 2), ultrasound (*n* = 2), radiographs (*n* = 1), physical examination (*n* = 1), skin culture (*n* = 1), and wound care (*n* = 1). When asked if sedation or anesthesia was used for the procedures (excluding euthanasia), six (75 percent) of eight respondents selected “yes,” two selected “no,” and zero respondents selected “unsure.”

When asked, “Does your institution use cooling or freezing for immobilization or euthanasia of GPOs?” 17 (81 percent) of 21 respondents selected “no,” four (19 percent) selected “yes, as a second step following chemical sedation/anesthesia,” and zero respondents selected “yes, as the sole means of immobilization” or “unsure.” Respondents who selected “yes” were prompted to describe the procedure, including temperatures and time frames. A respondent reported observing an animal that remained alive after sitting at 0 degrees Celsius for one to two hours. One respondent indicated that ice water is used with chemical methods, and two reported that freezing is used as a second step. When asked if the facility has euthanized GPOs, 16 (80 percent) out of 20 respondents selected “yes,” three respondents selected “no,” and one respondent selected “unsure.”

Table 5 lists responses to three questions in which respondents could select multiple agents used for sedation or anesthesia, agents or methods used for euthanasia, or agents or methods associated with negative experiences. Additional comments on magnesium chloride included considerations for osmotic disruption (gradual titration) and the use of sodium bicarbonate to increase pH to the enclosure level. The times of induction (28 min), immersion (45 min), and recovery (60 min) were listed with a magnesium chloride dosage of 33 g/L. The time to unresponsiveness or effectiveness of euthanasia with magnesium chloride ranged from 20 min to 2 hours. Comments specific to ethanol noted reports of repeated inking, escape behavior, arm curling, mucus response, and prolonged recovery from brief durations of high ethanol concentrations. Slower induction and recovery were reported with cold water. One respondent reported ethanol doses of 0.03 to 0.06 percent adequate for light sedation with a 15–20 min induction and 20–30 min recovery. Two respondents reported using a dose of two percent ethanol, including notes that the dose was adequate for gill biopsy, the GPO maintained voluntary respiration, the induction time ranged from five to 14 min, and there was a 10-min recovery at 10 degrees Celsius. One respondent commented that doses ranging from one to two percent are adequate for wound management, biopsy, and hemolymph collection with 2–6 min inductions and 20–30 min recoveries. A dose of three percent was noted to cause decreased ventilation but was adequate for hemolymph collection after a 20 min induction followed by a 45 min recovery.

Multiple respondents described euthanasia protocols that started with magnesium chloride, then added ethanol, followed by a secondary method. Respondents noted keeping the GPO in solution for a delayed time frame (45 min to four hours). Decerebration, freezing, and preceding necropsy were noted as secondary methods. Pentobarbital was noted as a secondary method. However, a respondent reported that a GPO continued to be slightly responsive 135 min after injecting 10 mL into the right brachial heart and 45 min after injecting an additional 10 mL into the left branchial heart. Both magnesium chloride and ethanol were used in combination with secondary methods for euthanasia. Multiple respondents described techniques using magnesium chloride as a first step, followed by ethanol once the GPO was no longer responsive, and a secondary method.

When asked how the respondent determines that the appropriate level of sedation, anesthesia, or euthanasia has been achieved, 16 respondents provided descriptions. A total of 10 (63 percent) respondents commented on the change in response to stimuli, including decreased lack of response, loss of flight, response to light, and arm withdrawal. Other noted observations include changes in respiration (five respondents), color changes (four respondents), partial to complete loss of arm movement (three respondents), decreased loss of sucker adhesion (three respondents), loss of cardiac activity with euthanasia (three respondents), loss of righting (two respondents), corneal reflex (one respondent), and loss of consciousness (one respondent).

## 4. Discussion

### 4.1. Response Rate

Due to the unknown distribution of emails from listservs, the total distribution and response rate are undetermined. In survey one, respondents from 52 institutions participated, though the total number of North American institutions that house cephalopods is not known. As of December 2021, there were 228 AZA institutions in North America [18]. This study was limited to North American institutions due to the restrictions of the current IRB. However, broadening the audience to include worldwide institutions may provide helpful information in the future. In the second survey, respondents from 35 participating institutions from the first survey were emailed a more in-depth survey. While there were 25 responses, three respondents only answered the first question, confirming that they are at least 18 years of age, which allowed respondents to view the rest of the online survey. Some respondents may have answered it to view questions on the survey and closed out to return later. The survey did not allow respondents to save answers, close the window, and return, so these were counted as separate responses. Thus, counting respondents that answered more than one question (22 respondents) in the survey, the response rate was 63 percent in Survey 2.

### 4.2. Survey One

Since at least 40 North American institutions house GPOs and two respondents indicated that their institution is planning to house the species in the future, we propose that the species continues to be a common animal on display at zoological facilities. While GPOs are frequently housed with smaller invertebrates such as anemones, sea stars, or urchins, they are not social species and are typically housed without other GPOs. When a male and female are placed together for potential mating, the individuals can become aggressive, and injury, including death, can result [19]. Fewer GPOs may be housed due to their large size and need for individual housing. The survey was conducted during the severe acute respiratory syndrome coronavirus two (SARS-CoV-2) pandemic, which may have affected the acquisition of new animals through collection, transportation, travel restrictions, financial, or other resource restraints. The low number of GPOs at facilities may also affect the ability of aquarists, veterinarians, or biologists to publish veterinary research studies or case reports on this species due to a low sample size at an institution. Since 60 percent of respondents reported that a GPO had been anesthetized or sedated at their institution, further investigation of anesthesia techniques could be beneficial to the veterinary care of this species.

In 2012, the Aquatic Invertebrate Taxon Advisory Group (AITAG) conducted a survey on GPO husbandry with 33 participating institutions [1]. Results in the AITAG survey are similar to those in this survey in regard to the type of life support system and habitat temperatures. In both surveys, most respondents selected that the institution has closed systems. Open systems accounted for 16 percent of responses in this survey and 13 percent in the AITAG survey. The feasibility of flow-through systems for facilities in general and specifically for GPOs may be dependent on the location of the institution. In both survey one and the AITAG survey, the mode was 10 degrees C for average temperature, with similar ranges reported. While some facilities may have the ability to use flow-through systems from a naturally cold water source, many facilities must use chillers to regulate enclosure temperature. Survey one and the AITAG survey reflect a wide range of exhibit sizes, and the minimum size of the exhibit enclosure in both surveys was 1893 L (500 gallons) [1].

### 4.3. Survey Two

While most participating institutions acquire GPOs commercially from collectors or through donations, no respondents selected “other” means, indicating that none of the GPOs were reported to have hatched in human care. Cephalopod culture is a common practice in research facilities. It is described in a limited number of species, such as cuttlefish (*Sepia officinalis*), loliginid squid (*Sepioteuthis lessoniana*), Mexican four-eyed octopus (*Octopus maya),* and common octopus (*Octopus vulgaris)* [20]. While the breeding of GPOs in human care has been successful, including the hatching of paralarvae, there has only been one documented report of a GPO successfully reared from hatch to sexual maturity [1,21,22]. Few respondents indicate the facility directly collects from the wild, which may be due to location, travel, finances, permit, or other regulatory restraints. GPOs can be caught as bycatch, and in Alaska, this is most common in Pacific cod (*Gadus macrocephalus*) pot fisheries [23,24,25].

In the 2012 AITAG survey, respondents noted that GPOs are fed at frequencies of more than once a day to twice a week, with a mode of three times per week [1]. The results from survey two were similar, with a range of twice daily to twice a week feeding frequency, and over half of the respondents indicated a GPO is fed between three and five times per week. Since the second survey allowed respondents to type in answers, many noted a range instead of a single number, with some adding information about supplemental or enrichment feeds. Feeding frequency likely varies due to changes in appetite with life stage or temperature and the participation of the GPO in interactive programs. Responses for the amount of food each feed and the type of food varied, which reflects a lack of consensus or established nutritional guidelines for GPOs in human care. One study examined octopus den middens to identify 69 prey species of North Pacific GPOs from Puget Sound to the Aleutian Islands [26]. Most individual GPOs were found to be generalists, with crustaceans and bivalves making up a large percentage of prey identified in the midden [26]. All respondents who listed dietary items listed at least three different food items, which supports dietary variation. While the second survey did not include a question about dietary vitamins or supplements with frozen–thawed fish, it is a topic that warrants investigation in the future.

In the 2012 AITAG survey, 91 percent of respondents reported a pH of 7.9–8.1 or 8.1–8.3 [1]. The results of the second survey are very similar and match the typical pH range of saltwater. The average salinity ranged between 26–28 and 33–35 ppt in the 2012 AITAG survey [1]. Similar responses were noted in the second survey, which is also consistent with the salinity of saltwater. Nitrate levels reported in the second survey (0–29 mg/dL) were similar to values reported in the AITAG survey, which ranged from less than 10 to 30–40 mg/dL [1]. Foam fractionation was the most frequently noted type of life support system equipment employed in both the AITAG 2012 survey and the second survey [1]. Water quality details, including temperature and life support systems, were asked in the first survey but not in the second survey to decrease the number of questions. However, the temperature is a critical consideration when looking at other water quality parameters, such as dissolved oxygen. The water change frequency and change percentage vary considerably depending on the type of system. While multiple respondents included temperature and life support system information, it is difficult to analyze this data. If the survey is repeated, including all parameters in one survey is recommended.

Less than half of the respondents indicated that GPOs undergo quarantine. While it is generally ideal to keep new acquisitions separate for observation and biosecurity, there is currently no disease risk-based length of time recommendation for GPOs. As GPOs are housed individually and often in closed systems, facilities may choose to place an animal directly in the exhibit habitat. The five respondents who reported that GPOs go through a quarantine period described the length as 30 days, which is consistent with the default recommended quarantine period across species [1].

At least two-thirds of respondents (n = 21) reported that GPOs are visually examined by veterinarians at entrance and necropsy. Preventative visual examinations are performed at less than half of the respondent institutions. The responses for transfer examinations may be lower as many facilities may not transfer their GPOs. The variability of responses regarding veterinary examination frequency may reflect several factors, such as whether a veterinarian is on staff at the facility, the level of veterinary involvement in invertebrate care at the facility, and the relatively short lifespan of octopuses. The survey question did not specify that visual examinations could include visual assessment of the GPO during site visits or routine rounds, as such, these may not have been counted by all respondents and visual examinations may be occurring more often than what was documented in the survey. While veterinary procedures involving GPOs may not be considered common, eight responding institutions reported procedures including biopsies, imaging, skin scrapes or cultures, physical examinations, and wound care. Since six respondents reported using sedation or anesthesia for these non-lethal procedures, increasing knowledge of safe and effective anesthesia techniques for this species is warranted. As branchitis is often diagnosed histologically, gill biopsies can be performed as a tool to monitor for gill parasites [6]. Most (80 percent) of respondents report that euthanasia on GPOs has been performed at their institution. Once an octopus reaches senescence, they may develop behavioral changes that decrease their quality of life, such as autophagy, traumatic injuries, and prolonged anorexia. While many institutions report performing euthanasia, multiple respondents noted that this is not a common practice. Further investigation into anesthesia techniques may help with the perception of euthanasia in this species.

Hypothermia has been used as a method of immobilization and euthanasia [1,11]. According to the AVMA Guidelines on Euthanasia regarding aquatic invertebrates, acceptable methods include anesthetic overdose using agents such as, but not limited to, immersion with magnesium salts, ethanol, and eugenol. Secondary methods, including immersion in 70 percent alcohol or 10 percent formalin, pithing, freezing, and boiling, are recommended when death is difficult to confirm [27]. Unacceptable methods include leaving the invertebrate in a water container without adequate aeration [27]. Thus, freezing as a sole method of euthanasia could be considered prolonged death and unacceptable, especially if aeration is not provided. No respondents reported that the institution uses cooling or freezing as the sole means of immobilization or euthanasia. As respondents noted, when using freezing as a secondary method, it is recommended to minimize water volume as much as possible to minimize freezing time. In addition to freezing, secondary methods listed by survey respondents included mechanical decerebration, pentobarbital, and potassium chloride. Two respondents indicated that they had negative experiences with pentobarbital in GPO. While pentobarbital is a common and effective drug used in euthanasia due to its effects on the central nervous system and lethal cardiac effects, effective dosages and injection sites can vary among taxa [28,29].

Magnesium chloride and ethanol were the two most common agents selected for both sedation or anesthesia and euthanasia, which is consistent with current cephalopod literature [1,5,7,9,11,14,30,31,32]. In a 2018 study, Butler-Struben et al. proved that both ethanol and magnesium chloride block afferent and efferent neural signals in cuttlefish (*Sepia bandensis*) and octopus (*Abdopus aculeatus, Octopus bocki*) and can be considered effective anesthetic agents [14]. Effective magnesium chloride dosages for sedation or anesthesia in other octopus species include a ratio of 1:3 to 1:1 (19–35 g/L in *A. aculeatus* and *O. bocki*) and 37.5 g/L in *O. vulgaris* [14,30]. In the second survey, it was described that a dose of 33 g/L resulted in a light plane of anesthesia, while a higher dose of 47 g/L produced surgical anesthesia. Magnesium chloride doses of 75 to 100 g/L are recommended for cephalopod euthanasia [1,5,30]. Dosages reported for euthanasia with magnesium chloride in the survey had a much greater range of 20 to 200 g/L. Effective ethanol doses for sedation or anesthesia in other octopus species include one to four percent in *A. aculeatus* and *O. bocki* and two percent in *O. vulgaris* [11,14]. In GPOs, Gleadall reported an incomplete induction with a dose of three percent ethanol at 10 degrees Celsius but a higher temperature (20–21 degrees Celsius) resulted in a light plane of anesthesia sufficient for brief surgery [9]. This supports the impact of cold water on the effectiveness of ethanol immersion. The current recommended dose range is 2 to 5 percent [1,31]. Survey respondents listed doses ranging from 0.03 to 3 percent, ranging from light sedation to a deeper level of sedation, with decreased ventilation observed. Variations in effective dosages may reflect degrees of debilitation prior to anesthesia or euthanasia.

While noted induction times ranged from 2 to 20 min and noted recovery times ranged from 10 to 45 min, respondents reported that these periods were lengthened in colder water. Ethanol doses of 5 to 10 percent are recommended for cephalopod euthanasia [1,5,31]. Survey respondents listed a greater range of 5 to 27 percent ethanol as necessary for euthanasia. The relationship between magnesium chloride or ethanol dosage and temperature, weight, life stage, and health status has not been fully investigated in GPOs. Due to notes of repeated inking and escape behavior with ethanol, magnesium chloride may be perceived as less stressful. Negative experiences were described with both magnesium chloride and ethanol, including inadequate sedation and inking. While inking can be a sign of stress, animals are also stressed with handling and environmental changes.

Inhalant anesthetics such as isoflurane and sevoflurane are used clinically for the anesthesia of terrestrial invertebrates [33,34,35]. It was reported that isoflurane was used for anesthesia by two respondents and for euthanasia by one respondent. In a 2014 study, Polese et al. described the gradual application of isoflurane to two percent in *O. vulgaris* [8]. While gradual administration of two percent was effective for most animals, rapid application of 2.5 percent was lethal [8]. Survey respondents did not provide an isoflurane dosage; however, induction was reported to be 10 min in duration with a 15 min recovery. Due to challenges in cold water with ethanol and the potential for osmotic stress with magnesium chloride, isoflurane may be a promising alternative that warrants further investigation. Sevoflurane was not listed by any respondents.

Other less commonly selected agents include magnesium sulfate, benzocaine, MS-222, and dexmedetomidine. While eugenol is commonly used for anesthesia in other aquatic invertebrates, including cephalopods, it was not selected by any respondents [5,9]. In a recent octopus anesthesia study, clove oil at 0.15 mL/L did not meet the study criteria for induction anesthesia, and *O. maya* exposed to clove oil had slower growth rates than all other experimental groups [7]. Magnesium sulfate was only selected once for use in sedation and euthanasia. Messenger et al. reported it to be slightly less effective than magnesium chloride, with a larger quantity needed (200 g/L for euthanasia) [30]. While dexmedetomidine was listed as being used in conjunction with other methods, the dose and route were not described. Benzocaine use was described for anesthesia and euthanasia with the same dosages reported by Barord and Christie [17]. While effective, benzocaine is not commonly stored in large quantities and can be difficult to mix into a solution, so it may not be a preferred method due to ease of use. MS-222 was selected by two respondents for use in anesthesia and one respondent for use in euthanasia. MS-222 at 500 mg/L had no effect on the afferent neural signal in a cuttlefish. However, the animal went into respiratory arrest and did not recover [14]. Gleadall described violent contractions of the mantle and ink release with MS-222 at 100 mg/L in a GPO and variable effectiveness in other cephalopod species [9]. Five respondents indicated that they had a negative experience with MS-222 and noted the agent was ineffective and resulted in behavioral signs of stress (inking, escape). While Telazol^®^ was not selected or described as a method currently used, one respondent did note a possible stress and/or pain response observed in an animal characterized by curling up for hours before recovering.

Physical parameters of anesthesia of cephalopods are described in the Guidelines for Care and Welfare of Cephalopods in Research to include decreased or no response to noxious stimuli, decreased ventilation, flashing, or paling from decreased chromatophore tone, decreased movement including arm tone or sucker adhesion, change in posture or loss of righting reflex, and no response to light [4]. While all these parameters were noted by respondents, they were noted inconsistently. Lack of response to noxious stimuli was the most common parameter noted by 63 percent of respondents. Still, the next most common observation was a change in respiration, noted by 31 percent of respondents. Stages of cephalopod anesthesia from normal (stage 0) to medullary collapse (stage IV) are also described [1,17]. In general, survey respondents had no major consensus on determining when an appropriate level of sedation, anesthesia, or euthanasia had been achieved.

There is growing international interest in cephalopod welfare. Birch et al. define sentience as “the capacity to have feelings” beyond the feeling of pain, including comfort, discomfort, joy, excitement, etc. [36]. The 2021 report presented an argument for sentience in octopods using criteria including the presence of nociceptors, integrative brain regions, connections between nociceptors and integrative brain regions, responses to local anesthetics or analgesics, motivational trade-offs, self-protective behaviors, associative learning, and behavioral changes in response to local anesthetics or analgesics when injured [36]. While not yet included in the United States Animal Welfare Act, cephalopod molluscs and decapod crustaceans are included in the United Kingdom’s Animal Welfare (Sentience) Act of 2022 [37,38]. Similar to welfare considerations for vertebrates in human care, evaluating and maintaining positive welfare throughout a GPO’s life, including end-of-life management, is warranted. A 2022 study comparing post-euthanasia arm tip tissue demonstrated differences between healthy and senescent GPOs. This suggests that proactive welfare management should begin earlier in senescence due to neural degeneration and the loss of epithelial tissue [39]. Additional studies of anesthesia and analgesia are critical to determining which techniques may best decrease pain, discomfort, and distress. Techniques determined to be ineffective, having significant side effects, or leading to prolonged death are not recommended.

This survey study is limited to respondents from institutions that participated and should not be generalized to all facilities with GPOs. As the number of individuals who care for GPOs is likely a small percentage of the general population in North America, a sampling bias was necessary. This survey specifically targeted veterinarians and animal care professionals who work at facilities with GPOs. At the same time, the first survey was open to all facilities that house cephalopods; only 12 institutions that do not currently house cephalopods participated. This may be due to assumptions that this survey is specifically targeting facilities that house GPOs, as supported by the survey title, including GPOs specifically. Response details may also vary depending on the role of the respondent as well as their level of experience working with GPOs.

In conclusion, GPOs remain a commonly housed species in North American zoological institutions and research facilities. While there appear to be some similarities in GPO anesthesia with other cephalopod species, such as the use of magnesium chloride and ethanol for anesthesia or euthanasia, questions remain about the anesthetic process in GPOs and what factors may determine an effective dose. Further areas for prospective investigation include a comparison of anesthesia methods and examining potential relationships between effective drug dosage and factors such as temperature, animal size, and life stage. Additional areas that warrant further investigation include nutritional requirements in human care, treatment efficacy for pre-senescent diseases in GPOs, and hemolymph parameters. As GPOs are an important species in education, research, and marine ecosystems, survey data may help guide GPO husbandry, veterinary care, and research.

## Figures and Tables

**Table 1 vetsci-10-00448-t001:** Life support and water quality response summary. *n* refers to the number of respondents that indicated each response.

Parameter	Range	Mode (*n*)	Median	Mean	Total (*n*)	Survey	Ideal Levels per AZA Manual
Life support system	open, closed, other	Closed (29)			37	1	
Habitat volume (liters)	1893–16,389	1893 (3), 5678 (3)	5405	6162	35	1	>1893 L
Temperature range (°Celsius)	5–14.4 degrees				35	1	6.0–12.0 °C
Average temperature (°Celsius)	8.8–14.4	10 (5)	10	10.4	15	1	
Average pH or range	7.4–8.3	8 (4)			19	2	8.0–8.3
Average salinity or range (ppt)	29–36	32 (5)			20	2	28–33 ppt
Nitrate (mg/dL)	0–29	≤10 (4)			19	2	0–19 mg/L
Dissolved oxygen (% saturation)	90–109	90–100 (3)			8	2	85–95 %
Water change percentage	3–100	20 (3), 100 (3)			19	2	
Water change frequency	Continuous—weekly	weekly (8)			19	2	

Note for Survey 2 responses: When respondents were given the option to list other parameters, measured parameters included temperature, bromine, phosphates, copper, and alkalinity.

**Table 2 vetsci-10-00448-t002:** Listed filtration methods and enclosure decor or materials. *n* refers to number of respondents that indicated each response.

Filtration Methods (*n*)	Enclosure/Decor (*n*)
Foam fractionator (11)	Rockwork (14)
UV sterilization (8)	Fiberglass (8)
Bioball/biotower (6)	Acrylic (6)
Sand filter (5)	Natural or artificial kelp (5)
Canister filter (4)	Concrete or cement (4)
Sock filter (4)	Cave area (2)
Unspecified mechanical (4)	Other invertebrates specified (2)
Moving bed (3)	Tunnel (2)
Bead filter (2)	Gravel substrate (2)
Ozone (2)	Algae specified (2)
Carbon filter (2)	PVC (1)
Pleated pool filter (1)	Cobblestone substrate (1)
	Natural stone substrate (1)
	Crushed coral substrate (1)
	Large shells (1)
	Lava rocks (1)
**Total Respondents**: 20	20

**Table 3 vetsci-10-00448-t003:** GPOs at the institution are visually examined by veterinarians at the following times.

Examination	Responses	Percentage
Entrance/arrival	15	71
Necropsy	14	67
Only when needed	11	52
Preventative	10	48
Prior to transfer	8	38
Other	3	14
**Respondents** (*n*)	21	

**Table 4 vetsci-10-00448-t004:** How often do veterinary visual or physical examinations occur?

Frequency	Responses	Percentage
<Once/year or never	9	43
Twice a month or more	5	24
Every 3 to 6 months	3	14
Every 7 months to annually	3	14
Every 1 to 2 months	2	10
**Respondents** (*n*)	21	

Note: One response included two responses, totaling 22 responses

**Table 5 vetsci-10-00448-t005:** Agents or methods used by institutions for sedation, anesthesia or euthanasia, and agents with a reported negative experience (which can include previous experiences not at the institution).

Agent	Sedation or Anesthesia	*n*	Euthanasia	*n*	Negative Experiences	*n*
Magnesium chloride (66–75 g/L stock)	33–47 g/L, titrated	9	20–200 g/L	13	Inadequate, inking	2
Ethanol (ethyl alcohol, vodka)	0.03–3.0%	6	5–27%	9	Inadequate, inking	3
Isoflurane (in a covered trash can)	Induction 10 min, Recovery 15 min	2		1		0
MS-222	100 mg/L	2		1	Ineffective, inking, escape behavior	5
Magnesium sulfate		1		1		0
Benzocaine	500–2000 mg/L	1	>2000 mg/L (difficult to mix into solution)	1		0
Sevoflurane		0		0		0
Eugenol		0		0		0
Other (dexmedetomidine)		1		1		0
Mechanical decerebration	NA	NA	N/A	4	Rapid cut on the midline at the level of eyes	0
Pentobarbital	NA	NA	20 mL is not effective	2	Ineffective	2
Potassium chloride	NA	NA		1		0
Other (freezing)	NA	NA	N/A	1	NA	NA
Other (unspecified euthanasia solution)	NA	NA		1	NA	NA
Cooling	NA	NA	N/A	NA	Prolonged	1
Other (Telazol^®^)	NA	NA	N/A	NA	Curled up, prolonged recovery	1
**Total Respondents**		13		17		9

## Data Availability

The data presented in this study are available on request from the corresponding author. The data are not publicly available due to ethical considerations of protecting the data and privacy of respondents.

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
