# Peer review of "Characterization of Current Husbandry and Veterinary Care Practices of the Giant Pacific Octopus (Enteroctopus dofleini) Using an Online Survey"

_vetsci, 2023, doi:10.3390/vetsci10070448_

Round 1

Reviewer 1 Report

Minor changes noting in PDF. Authors should consider moving some of the "results" presented in the discussion into the results section. This would make the discussion less wordy and more readable.

Author Response

Hello,

Thank you for reviewing our article. 

Minor changes noting in PDF. Authors should consider moving some of the "results" presented in the discussion into the results section. This would make the discussion less wordy and more readable.

Changes noted in the PDF were appreciated and made. A few redundant results were removed or moved into the results section. See updated draft for changes. 

Thanks again! 

Sincerely, 

Ashley Kirby

Reviewer 2 Report

A nice study on a little understood topic, I enjoyed reading it.

Generally numbers 1-9 are presented as text, not numerals (unless it is a unit [ie 1mm]). L19 should be "one" and "nine". I would suggest changing this throughout.

L20: Could you please provide the litres or m2, gallons are not generally used in scientific writing and many readers (outside the USA) will be unfamiliar with the unit.

L52: work-up

L59: please briefly explain what Branchitis is (do similarly for other specialist veterinary terms/conditions)

 Ethics: Can you please provide a little info re what was done with the e-mail addresses in terms of subsequent data storage etc. This is important in consideration of data protection. Also, did the project go through ethical review as part of the project approval process?

Overall this is an interesting paper. I would like to see a little more made of the implications of these varied practices and if any should be investigated further with a view to non-use. You note that there are some substantial signs of discomfort/distress related to some of the protocols. These could be important signs of welfare compromise and, given that octopus are one of the few invertebrates covered by animal welfare legislation around the world, I think more welfare discussion is needed. This could focus on duration/induction/behavioural evidence and the need for positive welfare throughout life and during end-of-life mgmt. You should also include some info re the sentience/welfare debate around octopuses in the intro. 

Author Response

Hello,

Thank you for taking the time to review our article! 

A nice study on a little understood topic, I enjoyed reading it. Thank you. 

Generally numbers 1-9 are presented as text, not numerals (unless it is a unit [ie 1mm]). L19 should be "one" and "nine". I would suggest changing this throughout. These edits were made. 

L20: Could you please provide the litres or m2, gallons are not generally used in scientific writing and many readers (outside the USA) will be unfamiliar with the unit. Yes, thanks for catching that. Values are now in liters. 

L52: work-up This was changed, thank you. 

L59: please briefly explain what Branchitis is (do similarly for other specialist veterinary terms/conditions) It is inflammation of the gills and was added. 

 Ethics: Can you please provide a little info re what was done with the e-mail addresses in terms of subsequent data storage etc. This is important in consideration of data protection. Also, did the project go through ethical review as part of the project approval process?

Yes, we went through NC State University’s IRB approval process. No one besides the research team will have access to the raw data. Identifiers will be removed one month following collection of all data and no individual responses or direct quotes will be shared. The ethics section was also edited (see draft). 

Overall this is an interesting paper. I would like to see a little more made of the implications of these varied practices and if any should be investigated further with a view to non-use. You note that there are some substantial signs of discomfort/distress related to some of the protocols. These could be important signs of welfare compromise and, given that octopus are one of the few invertebrates covered by animal welfare legislation around the world, I think more welfare discussion is needed. This could focus on duration/induction/behavioural evidence and the need for positive welfare throughout life and during end-of-life mgmt. You should also include some info re the sentience/welfare debate around octopuses in the intro. 

Great points. An additional paragraph (line 498) was added in the discussion on welfare including sentience. A brief nod to the sentience debate was included in the introduction as well. 

Thanks again. 

Sincerely, 

Ashley Kirby

Reviewer 3 Report

The content and quality of this paper are very good.  It provides novel information on GPOs that are beneficial to zoo and aquarium collections and provides a summary of knowledge that can be found in one peer-reviewed reference.

This paper was well written.  There were a few minor typos and word choices that could be corrected, but no problems were seen with the quality of English language.

Author Response

Hello,

The content and quality of this paper are very good.  It provides novel information on GPOs that are beneficial to zoo and aquarium collections and provides a summary of knowledge that can be found in one peer-reviewed reference.

Thank you for taking the time to review our article! 

This paper was well written.  There were a few minor typos and word choices that could be corrected, but no problems were seen with the quality of English language.

A number of edits were made in the updated draft. 

Thanks again! 

Sincerely, 

Ashley Kirby